# Mechanical Properties of Alkali-Activated Slag Fiber Composites Varying with Fiber Volume Fractions

**DOI:** 10.3390/ma15186444

**Published:** 2022-09-16

**Authors:** Hyeon-Jin Lim, Chang-Geun Cho, Jang-Yeol You, Jong-Jin Jeong

**Affiliations:** 1Department of Architectural Engineering, Chosun University, Gwangju 61452, Korea; 2Department of Architectural Engineering, Songwon University, Gwangju 61756, Korea

**Keywords:** alkali-activated slag, multiple microcrack, sodium sulfate, calcium hydroxide, high ductile behavior, polyvinyl alcohol fiber

## Abstract

The mechanical properties of alkali-activated slag fiber composites (ASFC) were investigated with varying volume fractions of PVA (Polyvinyl alcohol) fibers. Ground granulated blast furnace slag (GGBS) and alkali-activators were used as the main binders instead of cement, which emits a large amount of carbon dioxide during the manufacturing process. The measured slump flow of ASFC showed a high fluidity at a fiber content of 1.5 vol.% or less. The tensile, flexural, and shear strength of ASFC showed higher values as the amount of fiber increased. Compared to the existing high ductility fiber composites showing strain hardening behaviors with a fiber content of 2.0 vol.%, ASFC proved that it could exhibit high ductility characteristics due to multi-microcracks even at low fiber mixing rates of 1.0% and 1.25%. ASFC could be expected to lower the manufacturing cost with a low fiber content and provide improved workability with high fluidity. In addition, when manufacturing structural components using the developed ASFC, it is expected that the amount of fiber could be selected and used according to the required performance.

## 1. Introduction

The world is confronted by the problem of environmental destruction, global warming, and resource depletion due to continuous industrialization, and global warming caused by greenhouse gases is considered to be the most serious problem. Ordinary Portland cement emits more than 0.9 tons of carbon dioxide to manufacture one ton of cement. It accounts for more than 7% of emissions from all industries [1,2,3,4]. As an approach to reducing carbon dioxide from cement production, some methods can be adopted such as reducing carbon dioxide emissions during cement production [3], reducing the amount of cement used by using industrial by-products, and using alkali-activated slag instead of ordinary Portland cement.

Similar to ordinary Portland cement, alkali-activated ground granulated blast furnace slag (GGBS) also has hydraulic reaction characteristics so that hardened alkali-activated GGBS mortars have the mechanical characteristics of high strength at early and long-term ages, and are known to exhibit advantages such as a high resistance to chemical attack and freeze-thaw as well as a low carbonation rate [5,6,7]. In the construction industries, ordinary concrete has many advantages as one of most widely used materials. However, its brittle characteristics due to the low strain rate and the decrease in durability due to local cracking remains a problem awaiting a solution. As a solution to this problem, high-ductile fiber composites have been designed by micromechanics and has been manufactured by mixing chopped PVA fibers to a volume fraction of less than 2.0 vol.%. The composite exhibited a high ductile deformation with crack control under uniaxial tensile stress and showed multiple microcracks of less than 100 μm. This characteristic of high-ductile behavior of the fiber composite could be attractive for its use as a construction material [8,9,10,11]. Studies on fiber-reinforced alkali-activated slag composites have been investigated by several researchers, and it is known that they have similar tendencies of mechanical properties as fiber composites based on ordinary Portland cement [12,13,14,15]. When manufacturing high-ductility fiber-reinforced composites, the fiber in the constituent material is very expensive, and it is necessary to control the amount of fiber according to the required performance of the fiber composite. However, several studies have been conducted with a fixed fiber volume fraction in one mixture [16,17], and it is necessary to evaluate the mechanical properties according to the volume fraction of fibers.

In this study, the characteristics of alkali-activated slag fiber composites (ASFC) according to the volume fraction of PVA fibers were newly investigated. A slump-flow test was performed to investigate the fluidity of the ASFC, and a series of mechanical tests on compression, uniaxial tensile, bending, and direct shear were performed to evaluate the hardened mechanical properties of ASFC.

## 2. Materials and Mixing of Composition

### 2.1. Materials

In order to manufacture ASFC, slag and alkali-activators in powder form were used as binding materials. For the combination of alkali-activators, calcium hydroxide, which is a caustic alkali having an advantage in the developing of strength, was used as the main activator, and sodium sulfate, a non-silicate weak acid salt, was used as an auxiliary activator [18]. The chemical composition of the GGBS is shown in Table 1; the specific gravity of the GGBS was 2.93, the Blaine fineness of the GGBS was 4300 ${\mathrm{cm}}^{2}/g$, and the basicity of the GGBS was 1.81. For aggregates, silica sand with an average diameter of 100 μm and a specific gravity of 2.65 was mixed in to reduce the shrinkage of the fiber composite, to secure sufficient rigidity, and to reduce the fracture toughness to increase the occurrence of steady-state cracks [19,20]. The reinforcing fiber mixed in the ASFC was a polyvinyl alcohol (PVA) fiber, which has a tensile stress of 1600 MPa and a length of 12 mm, and its detailed properties are shown in Table 2. High-range water-reducing admixture (HRWRA) and viscosity modifying admixture (VMA) were used to obtain the appropriate viscosity and the fluidity of the fiber composites as well as the dispersibility of fibers within the binders. In addition, a small amount of antifoaming agent was added to minimize the amount of harmful air bubbles and to improve the strength of the hardened ASFC [21].

### 2.2. Mixing of Composition

In the current experiments, ASFC was composed of five mixtures. The water-to-binder ratio, the sand-to-binder ratio, and the fiber content are shown in Table 3. In order to evaluate the mechanical properties according to the fiber contents of ASFC, a standard specimen, ASFC000, had no fibers, and fiber contents of four mixtures were determined in ranges from 1.0 to 2.0 vol.%. The binder of the ASFC was composed of 89.5% blast furnace slag, 7.5% calcium hydroxide and 3% sodium sulfate as alkali activators to a total of 100%. Each mixture of ASFC had the same water–binder ratio of 0.338, which was determined from several experiments to optimize the fiber dispersibility, fluidity, and viscosity of fiber composites by adjusting HRWRA and VMA according to the PVA fiber contents. Each mixture of fiber composites was weighed in a weight ratio and then mixed in a vertical mixer. Solid components including slag, alkali-activator, and silica sand were put in a mixer and mixed for one to two minutes. Water was added and further mixed for one to two minutes. After PVA fibers were mixed in, HRWRA and VMA were additionally mixed in. The mixer was stopped at a point where the dispersion of fibers was optimized, and then a slump-flow test was carried out to measure the fluidity of the ASFC. After that, fresh ASFC mixtures were cast into each mold for compressive, uniaxial tensile, bending, and direct shear tests. Each mold was cured in air by covering the exposed areas with a sheet for 1 day, and then demolded and cured in water at 23 ± 3 °C for 28 days [22].

### 2.3. Slump-Flow Test

From current experiments, slump flows of fresh specimens are shown in Table 4 as measured in Figure 1 [23]. The fluidity of ASFC was over 700 mm at fiber contents of 1.0 to 1.5 vol.%, and slightly decreased to a value of 540 mm at a fiber content of 2.0 vol.%. As a result of using the minimum amounts of HRWRA and VMA for fiber dispersibility in the current experiments, it seemed that the fluidity of ASFC mixtures could be appropriately controlled by adjusting the amounts of admixtures.

## 3. Experimental Investigation

### Mechanical Tests

To investigate the mechanical properties of the ASFC mixtures, experiments such as compressive, uniaxial tensile, bending, and direct shear tests were carried out. A series of test for compressive strength were conducted, as shown in Figure 2, after 28 days of curing by manufacturing a cube specimen with a size of 50 × 50 × 50 mm according to ASTM C109-07 [22]. Direct uniaxial tensile tests were carried out using a displacement control method at a speed of 0.2 mm/min after fabricating a dog-bone specimen proposed by Kim et al., as shown Figure 3 [24]. The tensile stress of the specimen was measured using a load cell on the top of the specimen, and two LVDTs (linear variable differential transducers) were attached to both sides of the specimen to measure the strain based on a 150 mm measurement distance. As shown in Figure 4, a series of panel-bending tests were also conducted by a four-point loading method using a panel specimen of 400 × 100 × 10 mm. Two rollers were loaded at a speed of 0.06 ± 0.04 MPa/s at a distance of 380 mm for the span and 50 mm from the center of the specimen. In the direct shear tests, as shown in Figure 5, the shear failure of a specimen was induced by placing a 5 mm notch on both sides of the specimen, and the shear stress was measured for an area of 50 mm × 80 mm by applying a load to the center of both ends of the specimen. From the bending test, the flexural stress of each specimen can be calculated using the Equation (1)
(1)$$f_{b}=\frac{3P\times l}{b\times h^{2}}$$
where $f_{b}$ is the flexural stress, P is the transverse load, l is the distance between the loading points and the support, and b and h are the width and height of the specimen, respectively.

## 4. Experimental Results and Discussions

### 4.1. Compressive Strength

The uniaxial compressive strength measured through cube specimens of ASFC mixtures was obtained as shown in Table 5. The average compressive strength of the ASFC specimens at ages of 28 days was measured to be 30.95~35.50 MPa. As for the test results of the compressive strength of the ASFC, the specimens mixed with fibers showed higher results than the specimens mixed without fibers. The specimen mixed with PVA fibers of 1.25 vol.% (ASFC125), which had the highest compressive strength, was measured to be higher than the ASFC150 and ASFC200, which had higher fiber contents. The reason is that an appropriate amount of fiber has a reinforcing effect, which improves crack control, support, and ductility when the specimen is under compressive strain [25,26,27]. However, if the fiber content is increased, the defect increases due to the initial voids formed by inappropriate initial microcracks at the interface between the fibers and the matrix, leading to a decrease in the bearing capacity [28,29].

### 4.2. Direct Uniaxial Tensile Behaviors

A series of direct uniaxial tensile tests were carried out for the five mixtures, and the tensile stress and strain curves measured by the load cell and LVDT are shown in Figure 6. Table 6 shows the average values of initial crack stresses, tensile stresses, and tensile strains. Each specimen mixed with PVA fibers showed strain-hardening behavior due to multi-microcracks, and all the specimens showed a high tensile strain of over 2.0 %. Such a result can be explained by the stress performance index, which is the ratio of the initial crack stress to the tensile stress. As the stress performance index increases, the occurrence of multiple cracks and the tensile deformation performance may increase [30]. The ASFC mixtures showed enhanced tensile performances in the initial crack strength, and the peak tensile stress and high ductile tensile strain were increased as the fiber contents increased. In the case of ASFC150, the average tensile strain was 4.93%, showing the highest result. Figure 7 and Figure 8 show the crack distribution for each specimen after the direct uniaxial tensile test, respectively. It could be seen that all mixtures except ASFC000 showed the formation of multi-microcracks at regular intervals. In addition, it could be seen that ASFC150, which had the highest tensile strain, had a narrow crack interval and a large number of cracks in the specimen.

### 4.3. Flexural Performances

The results of a series of panel bending tests according to the fiber contents of the ASFC mixtures are shown in Figure 9, and the measured flexural stresses and displacements of the specimens are shown in Table 7. The flexural stress of the standard specimen was 5.58 MPa and the corresponding displacement was 1.04 mm. The flexural stresses of the ASFC mixed with fibers were from 8.11 to 12.29 MPa, and the corresponding displacements were measured to be from 46.97 to 72.25 mm. All the specimens mixed with fibers showed multiple cracks and strain hardening behaviors, in which the flexural stresses and corresponding displacements increased after initial cracks occurred. The flexural stress of ASFC200 was 12.29 MPa and the corresponding displacement was 72.25 mm, which was about 133% and 69 times higher than the values of ASFC000. The flexural stress was higher than that of the uniaxial tensile stress, being 173.6% higher in the case of ASFC150. Figure 10 shows the multiple cracks with deflected shapes of the panel specimens after the flexural bending tests. The appearance of the lower part of each panel where multiple cracks occurred after the bending test is presented in Figure 11. It was observed from the panel bending test that all the specimens with fibers exhibited a strain hardening behavior after the cracks occurred.

### 4.4. Direct Shear Strength Test

Table 8 showed that from a series of direct shear strength tests, the measured shear strength increased as the fiber content of the ASFC mixtures increased. The shear stresses of the specimen of ASFC200 and the standard specimen, which had no fibers, reached failure at 7.04 MPa and 3.57 MPa, respectively, with a difference of about 97%. As a result, it was found that PVA fibers mixed in the ASFC specimens contributed to improving the shear strength, and the shear stress at failure increased according to the fiber contents. Figure 12 shows the final state of failure of each shear specimen.

## 5. Conclusions

In this study, ASFC mixtures were manufactured, in which cement was not used, and the mechanical properties of the mixtures were evaluated by conducting compressive, tensile, bending, and shear tests. GGBS and alkali-activators, calcium hydroxide and sodium sulfate, were used as binders for the ASFC mixtures, and PVA fibers were used as reinforcing materials with fiber contents from 1.0 to 2.0 vol.%, and the effect of the fiber contents of the mixtures was evaluated to compare with a reference specimen, which was mixed without fibers.

In the ASFC mixtures, the fibers were easily dispersed with a water–binder ratio of 0.338 with fiber contents from 1.0 to 2.0 vol.%, and the mixtures showed a high fluidity of up to 770 mm in the slump-flow test. In order to exhibit high fluidity at a fiber content of 2.0 vol.%, it is necessary to increase the water–binder ratio.The compressive strength of the ASFC increased from 5.3 to 14.7% due to fiber mixing, but there was no effect on increasing the compressive strength according to the increase in the fiber mixing rate. In addition, it is known that the specimens exhibit superior post-peak toughness than the specimen without fibers due to the bridging effect of the fibers under compression, so that a high toughness can be expected when applied to structural components [28,31].As the fiber mixing rate of the ASFC increased, the tensile, flexural, and shear strengths gradually increased compared to that of the reference specimen, showing the effect of increasing the fiber mixing, especially in the tensile test of ASFC with a fiber content of 1.0 vol.%, where the strength performance index was 1.28 and the tensile strain was 2.71%, and the bending and shear performance were 45% and 40% higher than that of the specimen with a fiber content of 0 vol.%, respectively. Through the experimental studies, it was proved that a high ductility could be exhibited even at a fiber mixing ratio of 1.0 vol.%.

From current experiments, it was found that the ASFC mixtures, in spite of no use of cement, could enhance the tensile, bending, and shear performances according to PVA fiber contents. As could been seen in high-ductile fiber composites, which were mixed with cement-based binders with a fiber content of 2.0 vol.% [16,17], ASFC could also exhibit a post-cracked strain hardening behavior due to multi-microcracks even at low fiber contents of 1.0 vol.% and 1.25 vol.%. Moreover, ASFC could be expected to give some satisfactory characteristics of fresh and hardened mixtures in terms of mixing with a lower fiber content, high fluidity, easy casting and compaction, and improved workability. In addition, it is expected that the cost of manufacturing structural components can be lowered by reducing the amount of expensive fibers used in ASFC.

## Figures and Tables

**Figure 1 materials-15-06444-f001:**
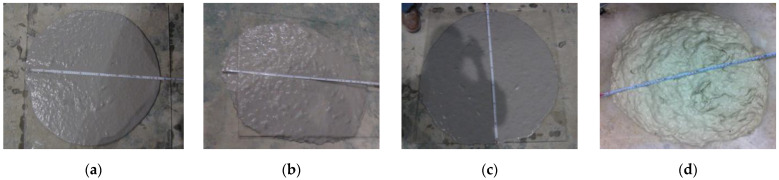
Slump-flow test: (**a**) ASFC100; (**b**) ASFC125; (**c**) ASFC150; (**d**) ASFC200.

**Figure 2 materials-15-06444-f002:**
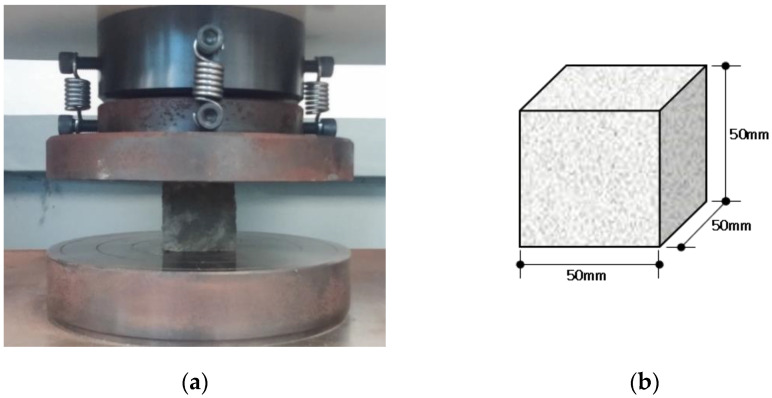
Slump-flow test: (**a**) Test setup; (**b**) Cube specimen.

**Figure 3 materials-15-06444-f003:**
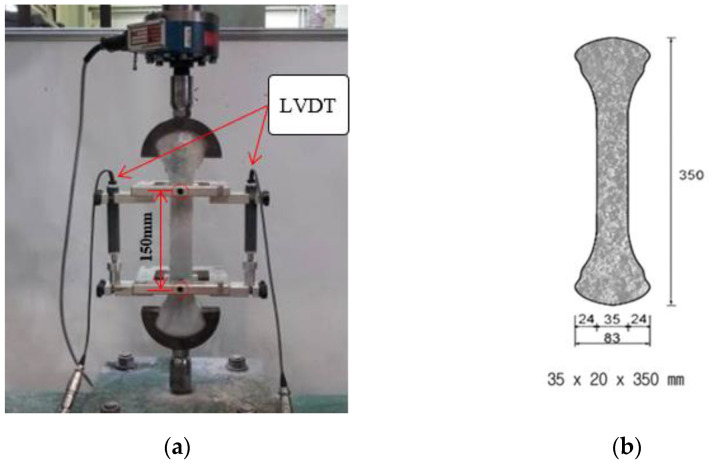
Direct uniaxial Tensile test: (**a**) Test setup; (**b**) Specimen geometry.

**Figure 4 materials-15-06444-f004:**
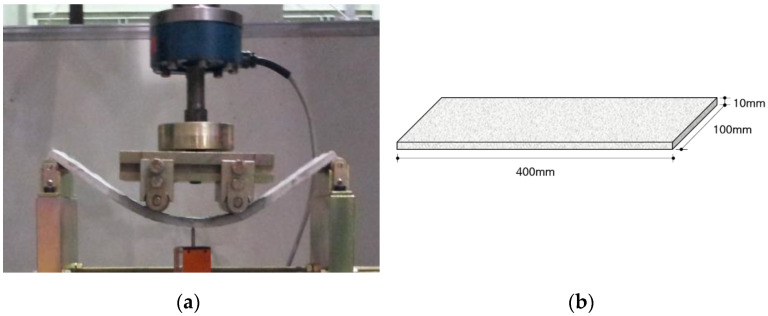
Bending test: (**a**) Test setup; (**b**) Specimen geometry.

**Figure 5 materials-15-06444-f005:**
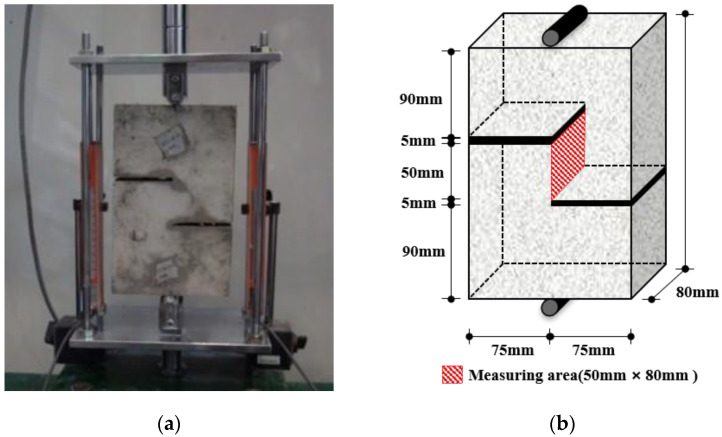
Direct shear test: (**a**) Test setup; (**b**) Specimen geometry.

**Figure 6 materials-15-06444-f006:**
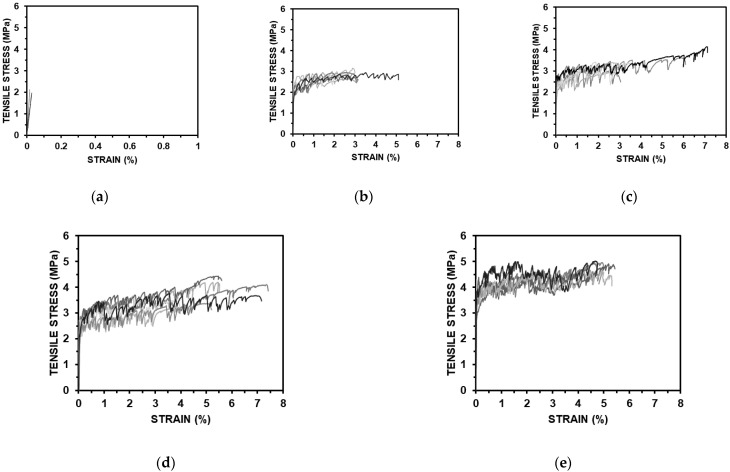
Direct uniaxial tensile stress–strain curves: (**a**) ASFC000 (PVA 0.00 vol.%); (**b**) ASFC100 (PVA 1.00 vol.%); (**c**) ASFC125 (PVA 1.25 vol.%); (**d**) ASFC150 (PVA 1.50 vol.%); (**e**) ASFC200 (PVA 2.00 vol.%).

**Figure 7 materials-15-06444-f007:**
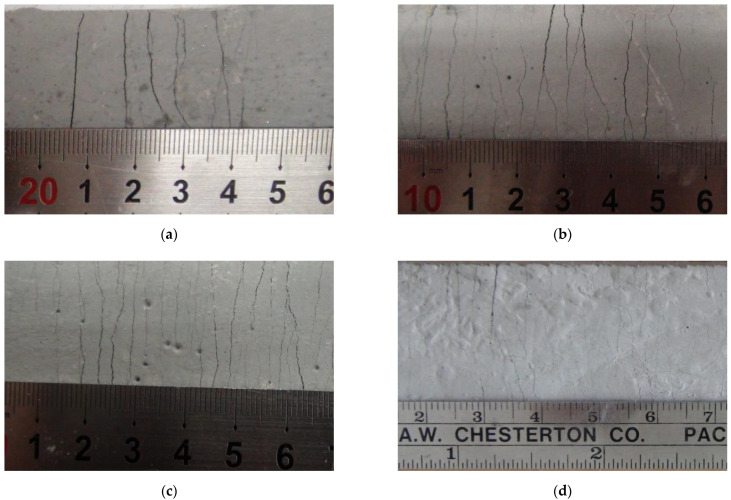
Multiple microcracks of tensile specimens: (**a**) ASFC100 (PVA 1.00 vol.%); (**b**) ASFC125 (PVA 1.25 vol.%); (**c**) ASFC150 (PVA 1.50 vol.%); (**d**) ASFC200 (PVA 2.00 vol.%).

**Figure 8 materials-15-06444-f008:**
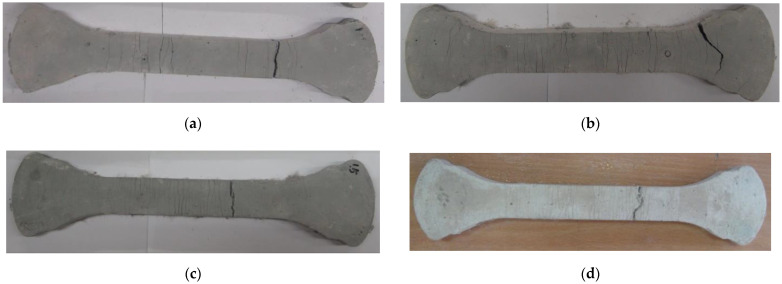
Multiple microcracks of tensile specimens: (**a**) ASFC100 (PVA 1.00 vol.%); (**b**) ASFC125 (PVA 1.25 vol.%); (**c**) ASFC150 (PVA 1.50 vol.%); (**d**) ASFC200 (PVA 2.00 vol.%).

**Figure 9 materials-15-06444-f009:**
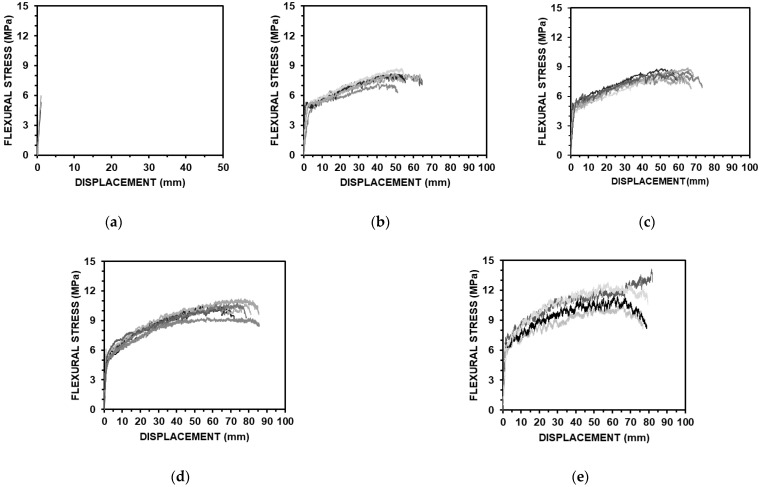
Flexural stress–displacement curves: (**a**) ASFC000 (PVA 0.00 vol.%); (**b**) ASFC100 (PVA 1.00 vol.%); (**c**) ASFC125 (PVA 1.25 vol.%); (**d**) ASFC150 (PVA 1.50 vol.%); (**e**) ASFC200 (PVA 2.00 vol.%).

**Figure 10 materials-15-06444-f010:**
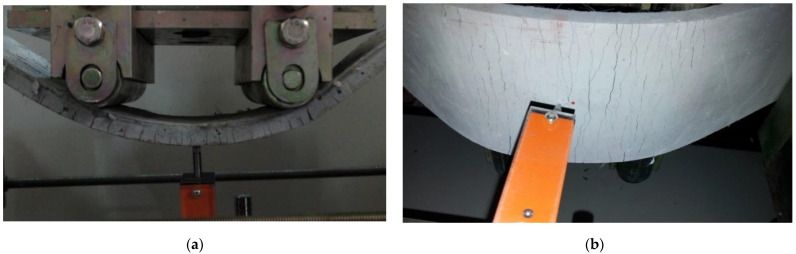
Multiple microcracks in bending test: (**a**) ASFC100 (PVA 1.00 vol.%); (**b**) ASFC150 (PVA 1.50 vol.%).

**Figure 11 materials-15-06444-f011:**
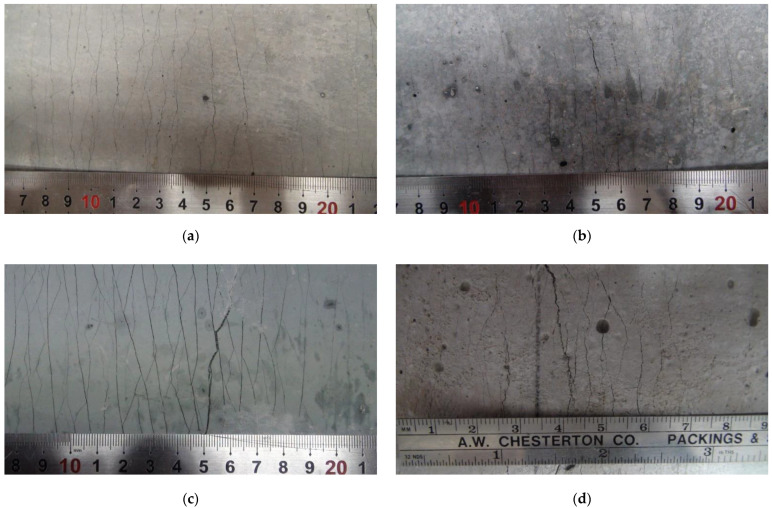
Multiple microcracks in bending panel: (**a**) ASFC100 (PVA 1.00 vol.%); (**b**) ASFC125 (PVA 1.25 vol.%); (**c**) ASFC150 (PVA 1.50 vol.%); (**d**) ASFC200 (PVA 2.00 vol.%).

**Figure 12 materials-15-06444-f012:**
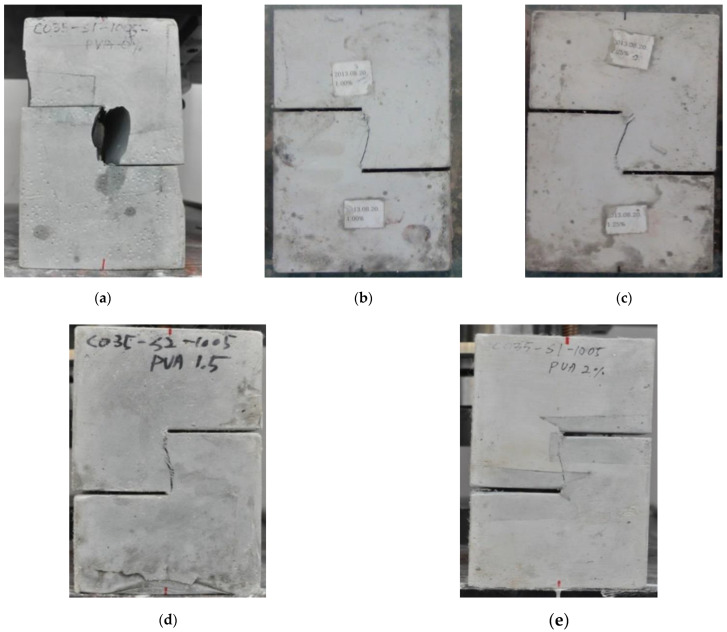
Failure of shear specimens: (**a**) ASFC000 (PVA 0.00 vol.%); (**b**) ASFC100 (PVA 1.00 vol.%); (**c**) ASFC125 (PVA 1.25 vol.%); (**d**) ASFC150 (PVA 1.50 vol.%); (**e**) ASFC200 (PVA 2.00 vol.%).

**Table 1 materials-15-06444-t001:** Chemical composition of GGBS.

Material	Chemical Composition (%)									
GGBS	SiO_2_	AI_2_O_3_	CaO	Fe_2_O_3_	SO_3_	MgO	K_2_O	Na_2_O	TiO_2_	Blaine (${\mathrm{cm}}^{2}/g$)
	31.57	13.57	43.26	0.38	4.53	4.86	0.41	0.18	0.55	4300

**Table 2 materials-15-06444-t002:** Properties of PVA (Polyvinyl alcohol) Fiber.

Diameter (mm)	Length (mm)	Tensile Strength (MPa)	Elongation (%)	Young’s Modulus (GPa)	Oil Content (%)
0.04	12	1600	6	37	0.8

**Table 3 materials-15-06444-t003:** Properties of PVA (Polyvinyl alcohol) fiber.

Mix ID	Binder			Water	Sand	VMA ^2^	HRWRA ^3^	PVA ^4^ (vol.%)
	GGBS ^1^	Alkali-Activator						
		Ca(OH)_2_	Na_2_SO_4_					
ASFC000	0.895	0.075	0.03	0.338	0.40	0.0007	0.005	0.00
ASFC100							0.012	1.00
ASFC125							0.015	1.25
ASFC150							0.018	1.50
ASFC200							0.020	2.00

Note: All numbers are mass ratios of binder weight except PVA fiber. ^1^ GGBS: Ground Granulate Blast-furnace Slag. ^2^ VMA: Ingredient (Methyl cellulose), pH (6.0). ^3^ HRWRA: Polycarboxylate superplasticizer, water reduction up to 30%. ^4^ PVA: Polyvinyl alcohol.

**Table 4 materials-15-06444-t004:** Slump-flow test results.

Mix ID	PVA Fiber (vol.%)	Slump-Flow (mm)
ASFC100	1.00	710
ASFC125	1.25	770
ASFC150	1.50	730
ASFC200	2.00	540

**Table 5 materials-15-06444-t005:** Uniaxial compressive strength.

Mixture ID	Fiber Contents (vol.%)	Compressive Strength (MPa)
ASFC000	0.00	30.95 ± 1.76
ASFC100	1.00	34.25 ± 2.73
ASFC125	1.25	35.50 ± 1.78
ASFC150	1.50	33.70 ± 2.14
ASFC200	2.00	32.60 ± 4.42

**Table 6 materials-15-06444-t006:** Direct uniaxial tensile test results.

Mixture ID	First Cracking Stress (MPa)	Tensile Stress (MPa)	Tensile Strain (%)	Stress Performance Index
ASFC000	2.06 ± 0.11	2.06 ± 0.11	0.02 ± 0.01	-
ASFC100	2.27 ± 0.24	2.90 ± 0.16	2.71 ± 0.40	1.28
ASFC125	2.60 ± 0.19	3.43 ± 0.40	3.89 ± 1.86	1.32
ASFC150	2.76 ± 0.26	3.80 ± 0.46	4.93 ± 1.32	1.38
ASFC200	3.86 ± 0.52	4.88 ± 0.16	4.68 ± 0.35	1.26

**Table 7 materials-15-06444-t007:** Bending test results.

Mixture ID	Flexural Stress (MPa)	Displacement (mm)
ASFC000	5.58 ± 0.45	1.04 ± 0.04
ASFC100	8.11 ± 0.51	47.97 ± 2.30
ASFC125	8.48 ± 0.34	56.45 ± 9.71
ASFC150	10.22 ± 0.76	65.51 ± 9.50
ASFC200	12.29 ± 1.55	72.25 ± 8.24

**Table 8 materials-15-06444-t008:** Results of shear strength test.

Mixture ID	Shear Strength (MPa)
ASFC000	3.57 ± 0.06
ASFC100	5.01 ± 0.36
ASFC125	5.75 ± 0.52
ASFC150	6.26 ± 0.73
ASFC200	7.04 ± 0.11

## Data Availability

Data sharing is not applicable to this article.

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
