# Peer review of "Mechanical Properties of Alkali-Activated Slag Fiber Composites Varying with Fiber Volume Fractions"

_materials, 2022, doi:10.3390/ma15186444_

Round 1
Reviewer 1 Report
The work presented by the authors is really interesting. The following are the observations and need to be addressed:
1. Reference section is weak. Less no. of references.
2. Table 3 is confusing. What are the values for Mix ID ASFC000, ASFC100, ASFC150 & ASFC200.
3. Figure 3 is not cited in the text
4. SEM analysis can be done to analyze the failures.
Author Response
Dear Reviewer 1
Thanks for your advice on this paper. Authors now submit a revised paper following on your advices (see an attached file).
Thanks

Reviewer 2 Report
Please see the attached file.

Author Response
Dear Reviewer 2
Thanks for your advice on this paper. Authors now submit a revised paper following on your advices (see an attached file).
Thanks

Round 2
Reviewer 2 Report
1.I previously gave several references to strength the citations [4-5] but it seems that the author cited more references with no relation to chemical attacks. The following references must be cited because none of the references [5-9] is about chemical attacks, Ren, J., L. Zhang, B. Walkley, J.R. Black, and R. San Nicolas, Degradation resistance of different cementitious materials to phosphoric acid attack at early stage. Cement and Concrete Research, 2022. 151: p. 106606; Ren, J., L. Zhang, Y. Zhu, Z. Li, and R. San Nicolas, A Comparative Study on the Degradation of Alkali-Activated Slag/Fly Ash and Cement-Based Mortars in Phosphoric Acid. Frontiers in Materials, 2022. 9; Fahim Huseien, G., J. Mirza, M. Ismail, S.K. Ghoshal, and A. Abdulameer Hussein, Geopolymer mortars as sustainable repair material: A comprehensive review. Renewable and Sustainable Energy Reviews, 2017. 80: p. 54-74.
2.What was the amount of HRWRA and VMA used? I didn't find the answer in the revised manuscript.
3. Section 4.1., why the compressive strength of specimens with higher fiber contents was not higher? You should give some mechanisms about this. You should give a dot-by-dot response to my previous comments.
Author Response
Dear Reviewer
Thanks for your review and interest on this article.
Now authors submit a revised paper following on your comments
as an attached file.
Best regards
